# Structural transition and re-emergence of iron's total electron spin in (Mg,Fe)O at ultrahigh pressure

Han Hsu [1✉] & Koichiro Umemoto[2]

Fe-bearing MgO [$(Mg_{1-x}Fe_x)O$] is considered a major constituent of terrestrial exoplanets. Crystallizing in the B1 structure in the Earth's lower mantle, $(Mg_{1-x}Fe_x)O$ undergoes a high-spin ($S = 2$) to low-spin ($S = 0$) transition at ~45 GPa, accompanied by anomalous changes of this mineral's physical properties, while the intermediate-spin ($S = 1$) state has not been observed. In this work, we investigate $(Mg_{1-x}Fe_x)O$ ($x \leq 0.25$) up to 1.8 TPa via first-principles calculations. Our calculations indicate that $(Mg_{1-x}Fe_x)O$ undergoes a simultaneous structural and spin transition at ~0.6 TPa, from the B1 phase low-spin state to the B2 phase intermediate-spin state, with Fe's total electron spin $S$ re-emerging from 0 to 1 at ultrahigh pressure. Upon further compression, an intermediate-to-low spin transition occurs in the B2 phase. Depending on the Fe concentration ($x$), metal–insulator transition and rhombohedral distortions can also occur in the B2 phase. These results suggest that Fe and spin transition may affect planetary interiors over a vast pressure range.

[1] Department of Physics, National Central University, Taoyuan City 320317, Taiwan. [2] Earth-Life Science Institute, Tokyo Institute of Technology, Tokyo 152-8550, Japan. ✉email: hanhsu@ncu.edu.tw

Fe-bearing MgO with the B1 (NaCl-type) structure, also known as ferropericlase $(Mg_{1-x}Fe_x)O$ (0.1 < x < 0.2), is the second most abundant mineral in the Earth's lower mantle (depth 660–2890 km, pressure range 23–135 GPa), constituting ~20 vol% of this region. Experiments and first-principles calculations have shown that B1 MgO remains stable up to ~0.5 TPa and transforms into the B2 (CsCl-type) structure upon further compression[1–14]. First-principles calculations have also predicted that B2 MgO remains dynamically stable up to at least ~4 TPa[12,15,16]. MgO has thus been considered a major constituent of terrestrial super-Earths (exoplanets with up to ~10 times of the Earth's mass), where the interior pressure can reach to the tera-Pascal regime[17,18].

With the abundance of Fe in the Earth interior, B1 MgO in the Earth's lower mantle contains 10–20 mol% of Fe. Likewise, in terrestrial super-Earths, MgO is expected to contain considerable amount of Fe. With the incorporation of Fe, physical properties of the host mineral can be drastically changed. For example, in B1 $(Mg_{1-x}Fe_x)O$, Fe undergoes a pressure-induced spin transition (also referred to as spin crossover) from the high-spin (HS, S = 2) to the low-spin (LS, S = 0) state at ~45 GPa[19–21]; the intermediate-spin (IS, S = 1) state has never been observed in experiments and has been ruled out by first-principles calculations[22]. The HS–LS transition of B1 $(Mg_{1-x}Fe_x)O$ is accompanied by anomalous changes of the structural, electronic, optical, magnetic, elastic, thermodynamic, and transport properties of this mineral[23–36]; it has also been suggested to change the iron diffusion and iron partitioning[21,23,37–39], to control the structure of the large low velocity provinces[40], and to generate the anti-correlation between bulk sound and shear velocities in the Earth's lower mantle[41]. Recently, seismological expression of the spin transition of B1 $(Mg_{1-x}Fe_x)O$ has also been reported[42]. Despite extensive studies on B1 $(Mg_{1-x}Fe_x)O$, B2 MgO, and the end member FeO (crystallizing in the B1 structure at pressure $P \lesssim 25$ GPa, undergoing complicated structural transitions upon compression[43–46], and stabilizing in the B2 structure at $P \gtrsim 250$ GPa[47,48]), effects of Fe and spin transition on the properties of B2 MgO and the B1–B2 transition remain unclear, especially for low Fe concentration (x ≤ 0.25) relevant to planetary interiors.

In this work, we study $(Mg_{1-x}Fe_x)O$ at ultrahigh pressure using the local density approximation + self-consistent Hubbard U ($LDA+U_{sc}$) method, with the Hubbard U parameters computed self-consistently. So far, $LDA+U_{sc}$ has been applied to various Fe-bearing minerals of geophysical and/or geochemical importance, including B1 $(Mg_{1-x}Fe_x)O$, Fe-bearing $MgSiO_3$ perovskite (bridgmanite) and post-perovskite, ferromagnesite $(Mg_{1-x}Fe_x)CO_3$, and the new hexagonal aluminous (NAL) phase[22,49–55]. Throughout these works, we have shown that spin-transition pressure determined by $LDA+U_{sc}$ is typically within 5–10 GPa around the experimental results, and the volume/elastic anomalies obtained by $LDA+U_{sc}$ are also in great agreement with experiments. With such accuracy, $LDA+U_{sc}$ has been established as a reliable approach to study Fe-bearing minerals at high pressure and is therefore adopted in this work. Further details of the computation and modeling are described in the Methods Section and Supplementary Information. To investigate the effects of Fe concentration, we perform calculations on $(Mg_{1-x}Fe_x)O$ with x = 0.125 and 0.25 using 16 and 8-atom supercells, respectively (Fig. 1). For B2 IS $(Mg_{0.75}Fe_{0.25})O$, we find ferromagnetic (FM) order more energetically favorable than antiferromagnetic (AFM) order; we therefore present the FM results in this paper.

## Results and discussion
**Self-consistent Hubbard U parameters**. Within $LDA+U_{sc}$, both the IS and LS states of B2 $(Mg_{1-x}Fe_x)O$ can be obtained at

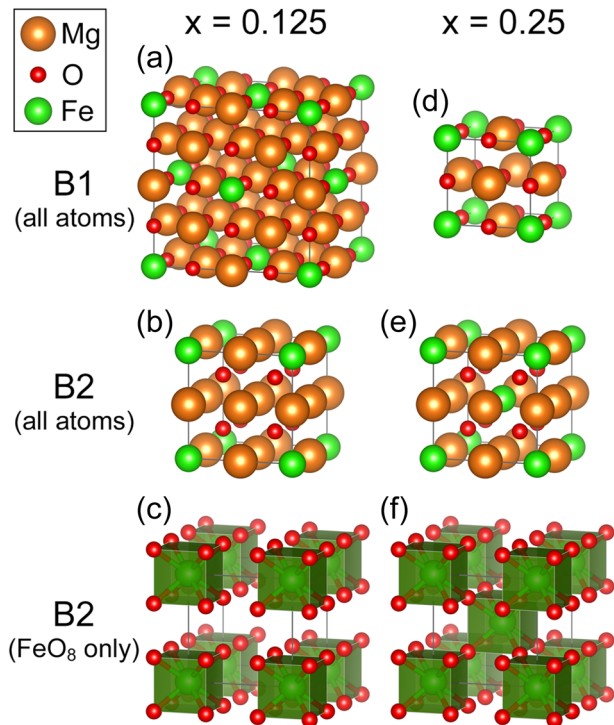

**Fig. 1 Supercells of B1 and B2 $(Mg_{1-x}Fe_x)O$.** For x = 0.125 (**a–c**) and x = 0.25 (**d–f**), 16- and 8-atom supercells are adopted, respectively. For the B2 phase, $FeO_8$ polyhedra are also plotted (**c**, **f**). In B2 $(Mg_{0.75}Fe_{0.25})O$, a 3D network of corner-sharing $FeO_8$ cubes is formed (**f**).

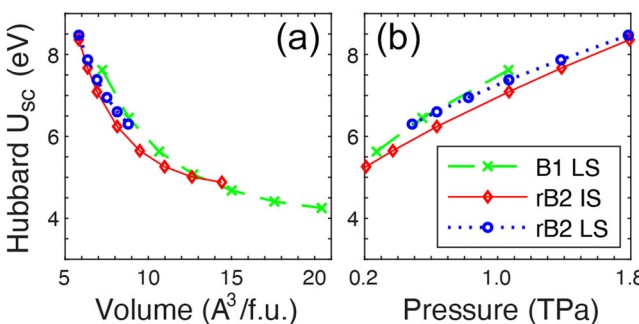

**Fig. 2 Self-consistent Hubbard U ($U_{sc}$) of Fe at ultrahigh pressure. a** $U_{sc}$ of intermediate-spin (IS) and low-spin (LS) Fe in B1 and rB2 $(Mg_{0.875}Fe_{0.125})O$ at various volumes. **b** $U_{sc}$ plotted with respect to pressure in the region of 0.2–1.8 TPa.

ultrahigh pressure, while the HS state can only be obtained at $P \lesssim 0.29$ TPa. The Hubbard $U_{sc}$ of Fe in $(Mg_{0.875}Fe_{0.125})O$ at various volume/pressure are shown in Fig. 2. Note that B2 $(Mg_{0.875}Fe_{0.125})O$ is stabilized via rhombohedral distortion (as further discussed in Figs. 3 and 4), hence referred to as rB2 hereafter. At ultrahigh pressure (0.5 < P < 1.8 TPa), as shown in Fig. 2b, $U_{sc}$ is mainly affected by pressure (increasing with P by ~2.5 eV) and marginally affected by the Fe spin state and crystal structure (by ~0.5 eV). In contrast, at P < 0.15 TPa, Fe spin/valence state affects $U_{sc}$ by up to ~2 eV, while pressure affects $U_{sc}$ by up to ~0.5 eV[22,49–54].

**Soft phonons in cubic B2 $(Mg_{0.875}Fe_{0.125})O$.** Fe spin states in B1 and B2 $(Mg_{1-x}Fe_x)O$ exhibit distinct properties, due to the host minerals' distinct crystal structures. In the B1 phase, Fe substitutes Mg in the 6-coordinate octahedral site (Fig. 1a, d), forming $FeO_6$ octahedra (Supplementary Fig. 1a and Note 1). In

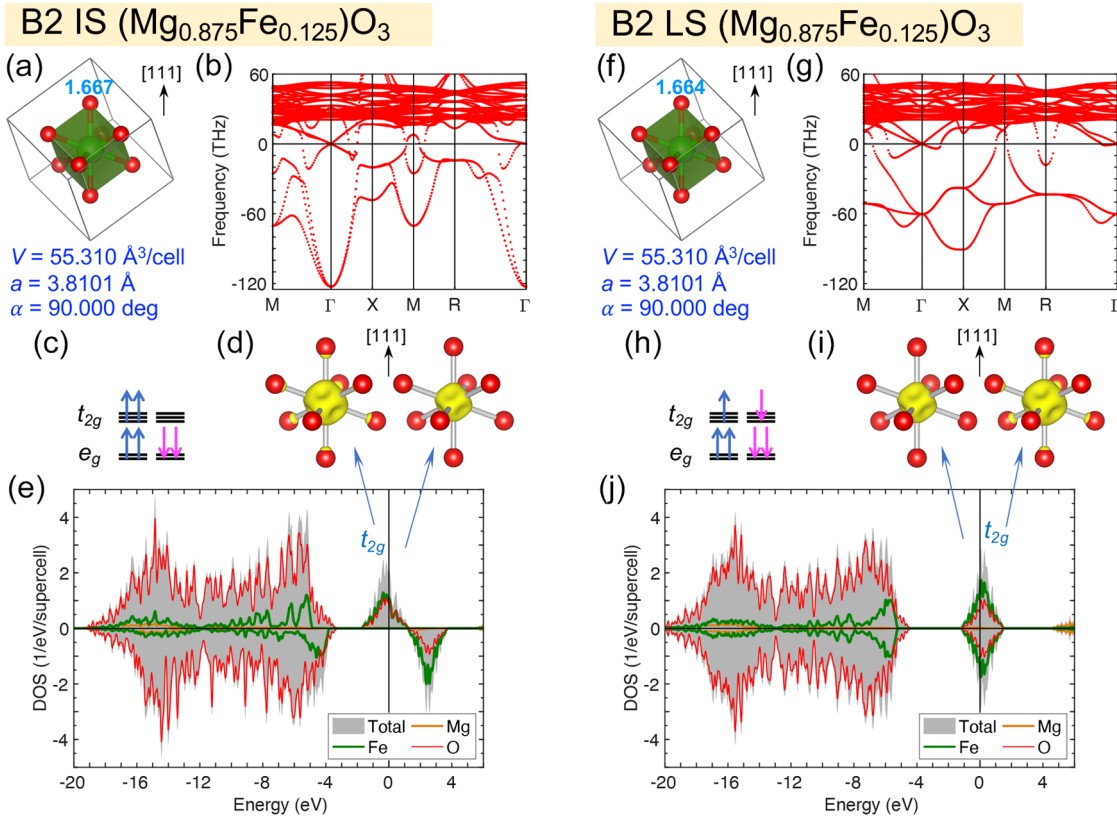

**Fig. 3 Atomic structure, stability, and electronic structure of cubic B2 (Mg$_{0.875}$Fe$_{0.125}$)O.** Here, the intermediate-spin (IS, **a–e**) and low-spin (LS, **f–j**) states at volume $V = 55.310$ Å$^3$/cell (6.914 Å$^3$/f.u., pressure $P \approx 1.07$ TPa) are shown. **a, f** FeO$_8$ cubes in the supercell, with the Fe-O bond lengths (in Å) indicated by the numbers next to the oxygen atoms, and the lattice parameters ($a$ and $\alpha$) listed below; **b, g** phonon dispersion; **c, h** orbital occupation; **d, i** integrated local density of states (ILDOS) of the filled and empty $t_{2g}$ states; **e, j** total and projected DOS, with the Fermi energy set as the reference (0 eV). In panels **a**, **d**, **f**, and **i**, the [111] direction is pointing upward.

FeO$_6$ octahedra, the $t_{2g}$ orbitals have lower energy than the $e_g$ orbitals; the orbital configurations of HS and LS Fe$^{2+}$ are $t_{2g}^4 e_g^2$ and $t_{2g}^6 e_g^0$, respectively (Supplementary Fig. 1b and Note 1). In the B2 phase, Fe substitutes Mg in the 8-coordinate site (Fig. 1b, e), forming FeO$_8$ polyhedra (Fig. 1c, f). For cubic B2 (Mg$_{1-x}$Fe$_x$)O with $x = 0.125$, after structural optimization, FeO$_8$ polyhedra remain cubic ($O_h$ symmetry), with all eight Fe-O bonds in the same length. Expectedly, the Fe-O bonds of the IS state are slightly longer than those of the LS state (Fig. 3a, f). Remarkably, cubic B2 (Mg$_{0.875}$Fe$_{0.125}$)O is dynamically unstable, regardless of the Fe spin state, as indicated by the soft phonon modes (negative phonon frequencies) shown in Fig. 3b, g. Nevertheless, the electronic structure of cubic B2 (Mg$_{0.875}$Fe$_{0.125}$)O still provides valuable insights. In FeO$_8$ cubes, the $t_{2g}$ orbitals have higher energy than the $e_g$ orbitals, and the orbital configurations of IS and LS Fe$^{2+}$ are both $e_g^4 t_{2g}^2$ (Fig. 3c, h). For the IS state, the $e_g$ orbitals are fully occupied, while the $t_{2g}$ orbitals are partially occupied by two spin-up electrons (Fig. 3c). A partially filled $t_{2g}$ band in the spin-up channel is thus formed, spanning across the Fermi energy (set as the reference, 0 eV) from $-1.7$ to $1.3$ eV, as indicated by the density of states (DOS) shown in Fig. 3e. The $t_{2g}$ characteristic of the $t_{2g}$ band can be visualized via the integrated local density of states (ILDOS) over the energy intervals $-1.7 < E < 0$ and $0 < E < 1.3$ eV for the filled and empty $t_{2g}$ states, respectively (Fig. 3d). For the LS state, the $t_{2g}$ orbitals are partially occupied by one spin-up and one spin-down electron (Fig. 3h), forming a partially filled $t_{2g}$ band in the interval of $-1.2 < E < 1.6$ eV (Fig. 3j). The ILDOS of the filled and empty $t_{2g}$ states (Fig. 3i) resemble those of the IS state (Fig. 3d). For both spin states, the completely filled $e_g$ bands are embedded in the oxygen band spanning over $-20 \lesssim E \lesssim -4$ eV (Fig. 3e, j). Evidently, the partially filled $t_{2g}$ band and thus the metallicity of cubic B2 (Mg$_{0.875}$Fe$_{0.125}$)O are the direct consequences of the cubic symmetry.

**Rhombohedrally distorted rB2 (Mg$_{0.875}$Fe$_{0.125}$)O.** Depending on the Fe spin state, dynamically unstable cubic B2 (Mg$_{0.875}$Fe$_{0.125}$)O is stabilized via rhombohedral compression or elongation. As shown in Fig. 4a/f, the IS/LS state is rhombohedrally compressed/elongated, with shortened/stretched Fe–O bonds along the [111] direction and rhombohedral angle $\alpha$ larger/smaller than 90°. The resultant rB2 structures for both spin states are dynamically stable with no soft phonon mode (Fig. 4b, g). With rhombohedral distortion, the FeO$_8$ cubes become FeO$_8$ dodecahedra ($D_{3d}$ symmetry), and the three $t_{2g}$ orbitals split into a singlet ($a_{1g}$), which is a $d_{z^2}$-like orbital along the [111] direction, and a doublet ($e_g'$). For the IS state, with shortened Fe-O bonds along the [111] direction, the $a_{1g}$ orbital has higher energy than the $e_g'$ orbitals; the four spin-up electrons occupy the $e_g$ and $e_g'$ orbitals, and the two spin-down electrons occupy the $e_g$ orbitals (Fig. 4c). With the splitting of $t_{2g}$ orbitals, an energy gap (~0.3 eV) is opened between the $e_g'$ and $a_{1g}$ bands, as indicated by the DOS (Fig. 4e). The $e_g'$ and $a_{1g}$ characteristics of these bands can be visualized via the ILDOS (Fig. 4d). For the LS state, with stretched Fe-O bonds along the [111] direction, the $a_{1g}$ orbital has lower energy than the $e_g'$ orbitals (Fig. 4h); the three spin-up and three spin-down electrons fully occupy the $e_g$ and $a_{1g}$ orbitals, with the $e_g'$ orbitals left unoccupied, resulting in a gap (~2.0 eV) between

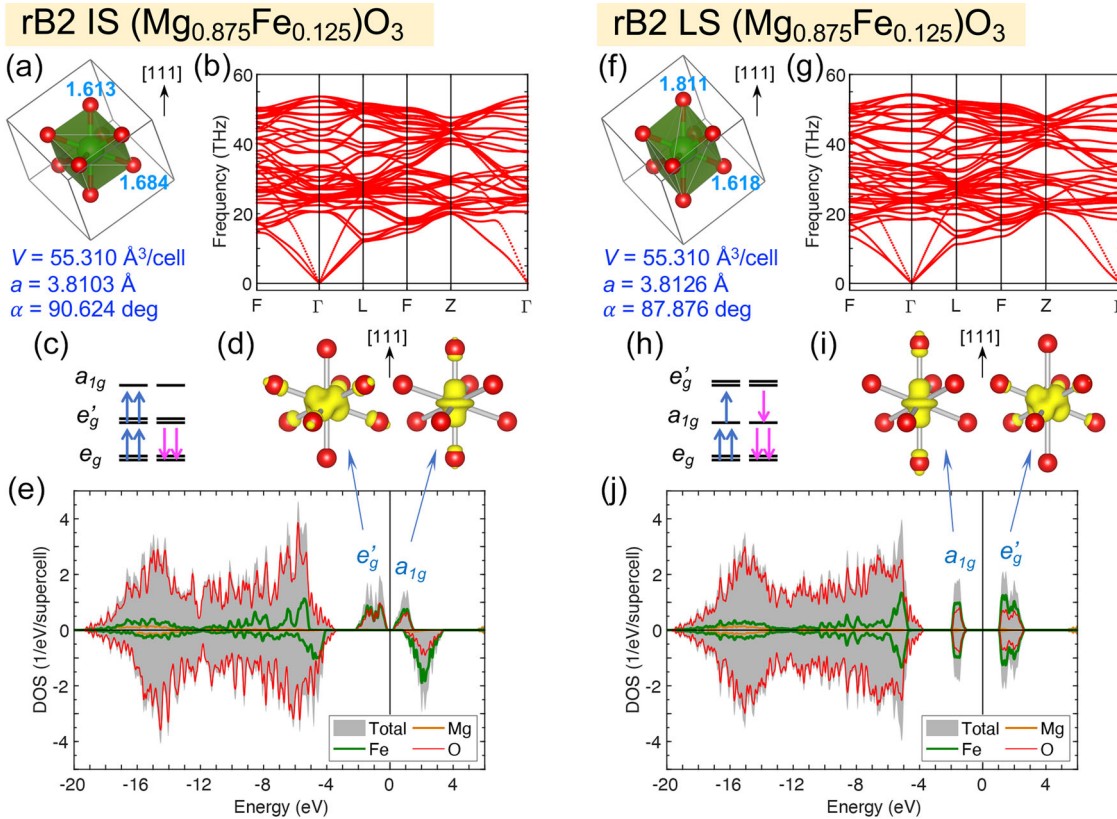

**Fig. 4 Atomic structure, stability, and electronic structure of rhombohedrally distorted rB2 (Mg$_{0.875}$Fe$_{0.125}$)O.** Here, the intermediate-spin (IS, **a–e**) and low-spin (LS, **f–j**) states at volume $V = 55.310$ Å$^3$/cell (6.914 Å$^3$/f.u., pressure $P \approx 1.07$ TPa) are shown. **a, f** FeO$_8$ dodecahedra in the supercell, with the Fe-O bond lengths (in Å) indicated by the numbers next to the oxygen atoms, and the lattice parameters ($a$ and $\alpha$) listed below; **b, g** phonon dispersion; **c, h** orbital occupation; **d, i** integrated local density of states (ILDOS) of the $e'$ and $a_{1g}$ bands; **e, j** total and projected DOS, with the Fermi level set as the reference (0 eV). In panels **a, d, f**, and (**i**), the [111] direction is pointing upward.

the $a_{1g}$ and $e'_g$ bands (Fig. 4j). The $a_{1g}$ and $e'_g$ characteristics of these two bands can also be visualized via the ILDOS (Fig. 4i). For both spin states, the completely filled $e_g$ bands are embedded in the oxygen bands spanning over $-20 \lesssim E \lesssim -4$ eV (Fig. 4e, j).

**Cubic B2 (Mg$_{0.75}$Fe$_{0.25}$)O remaining stable.** When the Fe concentration increases to $x = 0.25$, a three-dimensional (3D) network of corner-sharing FeO$_8$ cubes is formed in B2 (Mg0.75Fe0.25)O (Fig. 1f), while for $x \le 0.125$, the FeO$_8$ polyhedra are isolated/unconnected (Fig. 1c). For isolated FeO$_8$ polyhedra, rhombohedral distortion is allowed and favored, as observed in rB2 (Mg$_{0.875}$Fe$_{0.125}$)O (Fig. 4). In contrast, connectivity of the 3D FeO$_8$ network in B2 (Mg$_{0.75}$Fe$_{0.25}$)O suppresses the rhombohedral distortion and further stabilizes the cubic structure: Starting the structural optimization with rhombohedrally compressed/elongated rB2 IS/LS (Mg$_{0.75}$Fe$_{0.25}$)O, the crystal structure and FeO$_8$ polyhedra resume cubic symmetry within a few steps. Within LDA+$U_{sc}$, the LS state of cubic B2 (Mg$_{0.75}$Fe$_{0.25}$)O can be obtained throughout 0.2–1.8 TPa, while the IS state can only be obtained at $P < 1.1$ TPa and is subject to magnetic collapse: The total magnetization ($M$) decreases from $2\mu_B$/Fe to 0 in the region of $0.6 < P < 1.1$ TPa and vanishes at $P > 1.1$ TPa (Fig. 5a). (Note: For $x = 0.125$, the IS state retains $M = 2\mu_B$/Fe up to 1.8 TPa.) Regardless of the spin state and magnetization, cubic B2 (Mg$_{0.75}$Fe$_{0.25}$)O is dynamically stable with no soft phonon mode, even during the magnetic collapse (Fig. 5b–d). With FeO$_8$ cubes ($O_h$ symmetry), cubic B2 (Mg$_{0.75}$Fe$_{0.25}$)O and (Mg$_{0.875}$Fe$_{0.125}$)O have the same $3d$ orbital occupations and similar electronic structures. For cubic B2 IS (Mg$_{0.75}$Fe$_{0.25}$)O with $M = 2\mu_B$/Fe

(before the magnetic collapse), the spin-up $t_{2g}$ band spans across the Fermi energy (0 eV) while the spin-down $t_{2g}$ band lies above the Fermi energy (Fig. 5e), showing the same characteristic as cubic B2 IS (Mg$_{0.875}$Fe$_{0.125}$)O (Fig. 3e). Likewise, for cubic B2 LS (Mg$_{0.75}$Fe$_{0.25}$)O (Fig. 5g) and (Mg$_{0.875}$Fe$_{0.125}$)O (Fig. 3j), the $t_{2g}$ bands in both spin channels align, spanning across the Fermi energy. During the magnetic collapse ($0 < M < 2\mu_B$/Fe), the $t_{2g}$ bands in the spin-up and spin-down channels are shifted upward and downward, respectively (Fig. 5f).

**Complicated transitions of (Mg$_{1-x}$Fe$_x$)O.** To analyze the structural and spin transition of (Mg$_{1-x}$Fe$_x$)O, we compute the equations of state (EoS) of the B1 LS, (r)B2 IS, and (r)B2 LS states (Supplementary Fig. 2 and Note 2); the relative enthalpies ($\Delta H$) of these states with respect to the (r)B2 IS state are plotted in Fig. 6. For $x = 0.125$ (Fig. 6a), (Mg$_{0.875}$Fe$_{0.125}$)O transforms from the B1 LS state into the rB2 IS state at 0.642 TPa. Upon further compression, the rB2 IS state undergoes a spin transition to the rB2 LS state at 1.348 TPa. Throughout these transitions, (Mg$_{0.875}$Fe$_{0.125}$)O remains insulating (see Supplementary Fig. 1 and Note 1 for the insulating B1 LS state). Remarkably, in the simultaneous structural (B1–rB2) and spin (LS–IS) transition at 0.642 TPa, Fe's total electron spin $S$ re-emerges from 0 to 1, opposite to the perception that $S$ decreases upon compression. Furthermore, the IS state is energetically favorable over a wide pressure range (0.612–1.348 TPa), despite that IS state has never been observed in the Earth's lower-mantle minerals, including B1 (Mg$_{1-x}$Fe$_x$)O, Fe-bearing MgSiO$_3$ bridgmanite and post-perovskite, ferromagnesite (Mg$_{1-x}$Fe$_x$)CO$_3$, and the NAL

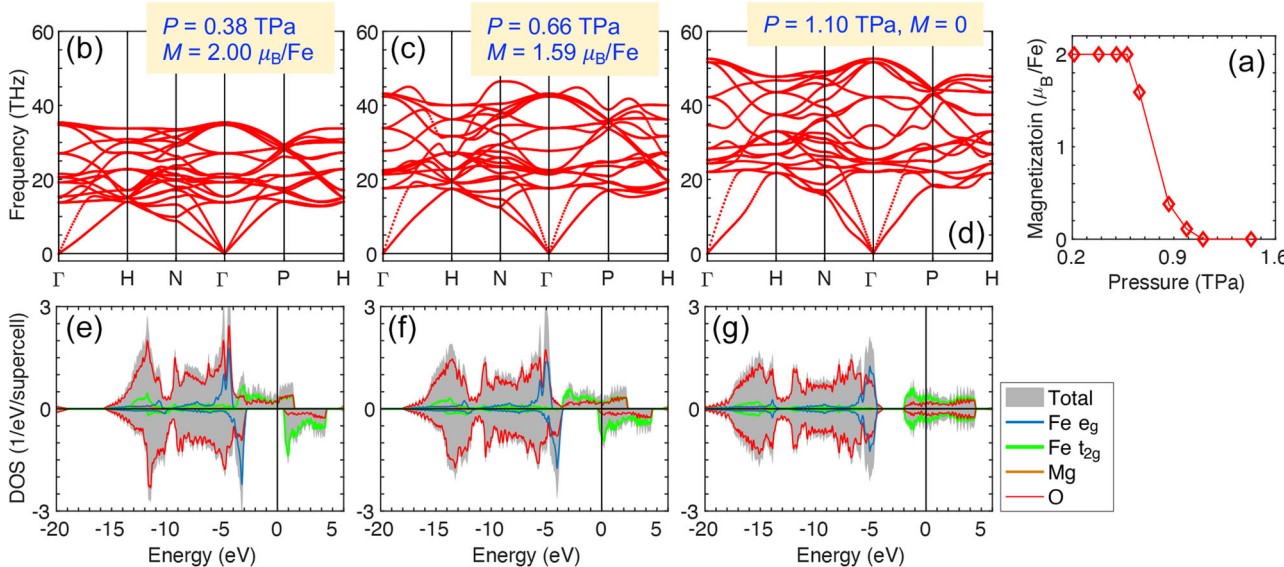

**Fig. 5 Stability, electronic structure, and magnetic collapse of cubic B2 intermediate-spin (IS) (Mg$_{0.75}$Fe$_{0.25}$)O. a** Total magnetization ($M$); **b**–**d** phonon dispersion; **e**–**g** total and projected density of states (DOS), with the Fermi energy set as the reference (0 eV). Panel (**a**) indicates that the IS state retains $M > 0$ at $P < 1.1$ TPa. At $P \geq 1.1$ TPa, only the nonmagnetic low-spin (LS) state ($M = 0$) can be obtained. Panels (**d**) and (**g**) thus also indicate the LS results.

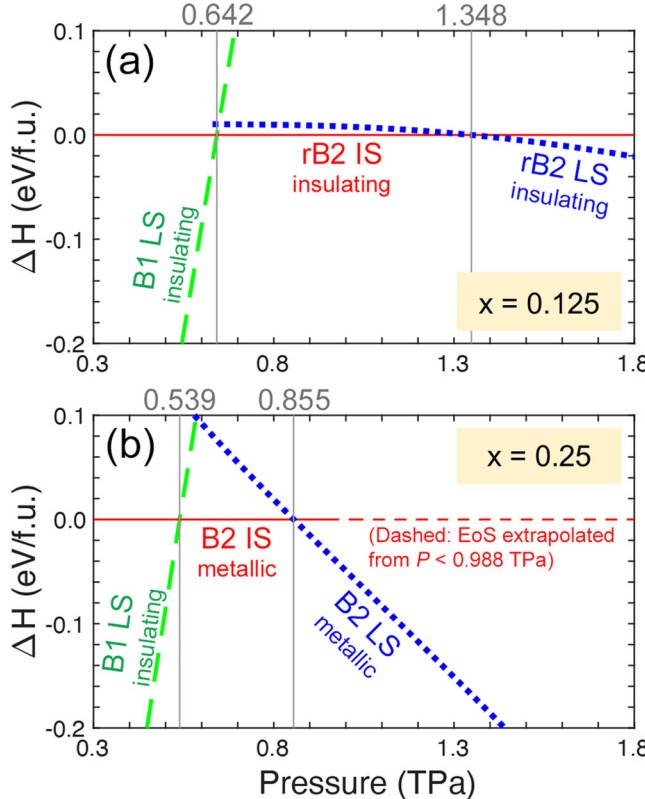

**Fig. 6 Relative enthalpies of (Mg$_{1-x}$Fe$_x$)O in various structural phases and spin states. a** $x = 0.125$, with the rB2 intermediate-spin (IS) state as the reference; **b** $x = 0.25$, with the B2 IS state as the reference. The vertical lines and the numbers above indicate the enthalpy crossings and transition pressures, respectively.

phase[22,49–54]. For $x = 0.25$ (Fig. 6b), a simultaneous structural, spin, and metal–insulator transition occurs at 0.539 TPa, from the insulating B1 LS state to the metallic B2 IS state (notice that $S$ increases). Upon further compression, an IS–LS transition occurs

in metallic B2 (Mg$_{0.75}$Fe$_{0.25}$)O at 0.855 TPa. From Fig. 6, effects of Fe concentration on the B1–(r)B2 transition pressure ($P_T^{B1/B2}$) can also be inferred. For Fe-free MgO ($x = 0$), we find $P_T^{B1/B2} = 0.535$ TPa (Supplementary Fig. 3 and Note 3), in agreement with other calculations[3–14]. As $x$ increases, $P_T^{B1/B2}$ first increases to 0.642 TPa at $x = 0.125$ (Fig. 6a) and then decreases to 0.539 TPa at $x = 0.25$ (Fig. 6b), indicating a trend of decreasing $P_T^{B1/B2}$ in the region of $0.125 \lesssim x \leq 1$, consistent with experiments: $P_T^{B1/B2}$ of FeO (~0.25 TPa)[47,48] is much lower than that of MgO (~0.5 TPa)[1–4]. [Note: (1) For $x = 0.25$ (Fig. 6b), the EoS of the B2 IS state is fitted using the data points at $P < 0.988$ TPa (where $M > 0$). (2) To examine the robustness of the LDA+$U_{sc}$ results shown in Fig. 6, we perform extensive test calculations using various methods. Similar results are obtained, as shown in Supplementary Figs. 4, 5, and Note 4].

In the Earth's mantle condition, Fe partitioning between B1 (Mg$_{1-x}$Fe$_x$)O, Fe-bearing MgSiO$_3$ bridgmanite, and post-perovskite varies with pressure, temperature, and even the Fe valence/spin state[21,23,37–39]. Likewise, in exoplanet interiors, Fe concentration in B2 (Mg$_{1-x}$Fe$_x$)O may vary with the depth or the interior region, due to the variation of Fe partitioning between B2 (Mg$_{1-x}$Fe$_x$)O and other minerals phases, including post-perovskite and/or high-pressure silicates[15,16]. Evident by comparing Fig. 6a and b, spin, structural, and metal–insulator transition of B2 (Mg$_{1-x}$Fe$_x$)O can also be induced by the change of Fe concentration ($x$). In the depth/region with pressure of 0.642–0.855 TPa, if $x$ increases from 0.125 to 0.25, a simultaneous structural and metal–insulator transition occurs [insulating rB2 IS (Mg$_{0.875}$Fe$_{0.125}$)O → metallic B2 IS (Mg$_{0.75}$Fe$_{0.25}$)O]; in the depth/region with pressure of 0.855–1.348 TPa, if $x$ increases from 0.125 to 0.25, a simultaneous structural, spin, and metal–insulator transition occurs [insulating rB2 IS (Mg$_{0.875}$Fe$_{0.125}$)O → metallic B2 LS (Mg$_{0.75}$Fe$_{0.25}$)O]. Even in the depth/region of $P > 1.348$ TPa, where only LS Fe$^{2+}$ exists, if $x$ increases from 0.125 to 0.25, a simultaneous structural and metal–insulator transition occurs [insulating rB2 LS (Mg$_{0.875}$Fe$_{0.125}$)O → metallic B2 LS (Mg$_{0.75}$Fe$_{0.25}$)O]. On the other hand, if $x$ decreases from 0.25 to 0.125, the aforementioned transitions would be reversed. Based

on the above analysis, metal–insulator transition is always included in the composition-induced transitions of the B2 phase, suggesting that variation of Fe partitioning can significantly change the electrical and thermal transport properties of exoplanet interiors.

**Implications of spin transition in the B2 phase**. At temperature $T \neq 0$, spin transition goes through a mixed-spin (MS) phase/state, in which different spin states coexist. For B2 $(Mg_{1-x}Fe_x)O$, only the IS and LS states are relevant. Within the thermodynamic model detailed in Supplementary Note 5, the LS fraction ($n_{LS}$) in the MS phase can be written $n_{LS} = 1/[1 + 3\exp(\Delta H/k_B Tx)]$, where $\Delta H \equiv H_{LS} - H_{IS}$, and the IS fraction $n_{IS} = 1 - n_{LS}$. Despite that lattice vibration is not considered, the results obtained from this approach have been shown in great agreement with room-temperature experiments[22,50,53,54,56]. In Fig. 7, the LS and IS fractions, compression curves, and bulk modulus of B2 $(Mg_{1-x}Fe_x)O$ at $T = 300$ K are shown. For $x = 0.125$, the IS–LS transition is smooth and spans over a wide pressure range (Fig. 7a), due to the small enthalpy difference ($\Delta H$) between the rB2 IS and LS states (Fig. 6a). The compression curves $V(P)$ of the MS, IS, and LS states are nearly the same (Fig. 7b); their difference is barely noticeable even by plotting the relative volume difference with respect to pure B2 MgO, namely, $(V - V_{MgO})/V_{MgO}$ (Fig. 7c). As a consequence, the bulk modulus $K \equiv -V\partial P/\partial V$ barely changes during the spin transition (Fig. 7d). For $x = 0.25$, the spin transition is more abrupt (Fig. 7e), due to the larger $\Delta H$ between the B2 IS and LS states (Fig. 6b). With increased Fe concentration, the volume difference between the LS and IS states increases (Fig. 7f), resulting in prominent volume reduction (~0.5%) (Fig. 7g) and bulk modulus softening (~22%) (Fig. 7h). While the full elastic tensor ($C_{ij}$) and shear modulus ($G$) are not computed, the volume and elastic anomalies shown in Fig. 7g and h clearly indicate anomalous softening of bulk sound velocity $v_\Phi = \sqrt{K/\rho}$ ($\rho$: density) and compressional wave velocity $v_P = \sqrt{(K + \frac{4}{3}G)/\rho}$. Furthermore, within the phonon gas model, lattice thermal conductivity $\kappa \approx (1/3)C_V v_P^2 \tau$ ($C_V$: heat capacity; $\tau$: average phonon scattering time)[57], suggesting that anomalous change of $v_P$ may play a role in the anomalous change of thermal conductivity. In the B1 phase, anomalous $v_\Phi$, $v_P$[20,28–33] and $\kappa$[35,36] have all been observed. The volume/elastic anomalies accompanying the spin transition of the B2 phase may thus be a possible source of seismic and thermal anomalies in exoplanet interiors.

Experiments and computations have confirmed that HS–LS transitions in B1 $(Mg_{1-x}Fe_x)O$ and ferromagnesite $(Mg_{1-x}Fe_x)CO_3$ are accompanied by anomalous changes of major thermal properties, including thermal expansivity, heat capacity, Grüneisen parameter, thermal conductivity, and thermoelasticity[20,28–33,35,36,55]. As the temperature increases, the spin-transition pressure increases, and the transition is broadened[29,55]. Remarkably, for $(Mg_{0.75}Fe_{0.25})CO_3$, the anomalous increase of heat capacity retains its magnitude (~12%) without smearing out, even at high temperature[55]. Likewise, for the B2 phase, anomalous changes of thermal properties accompanying the spin transition can be expected. To investigate such anomalies at high $P$–$T$ conditions, vibrational free energy must be included. Thermal calculations are thus highly desirable and will be left for future studies.

In summary, we have investigated $(Mg_{1-x}Fe_x)O$ ($x \leq 0.25$) at ultrahigh pressure up to 1.8 TPa via first-principles calculations. Our calculations indicate that Fe greatly affects the properties of $(Mg_{1-x}Fe_x)O$. For $x = 0.125$, insulating $(Mg_{0.875}Fe_{0.125})O$ undergoes a simultaneous structural and spin transition (B1 LS → rB2 IS) at 0.642 TPa, followed by a spin transition (rB2 IS–LS) at 1.348 TPa. For $x = 0.25$, $(Mg_{0.75}Fe_{0.25})O$ undergoes a simultaneous

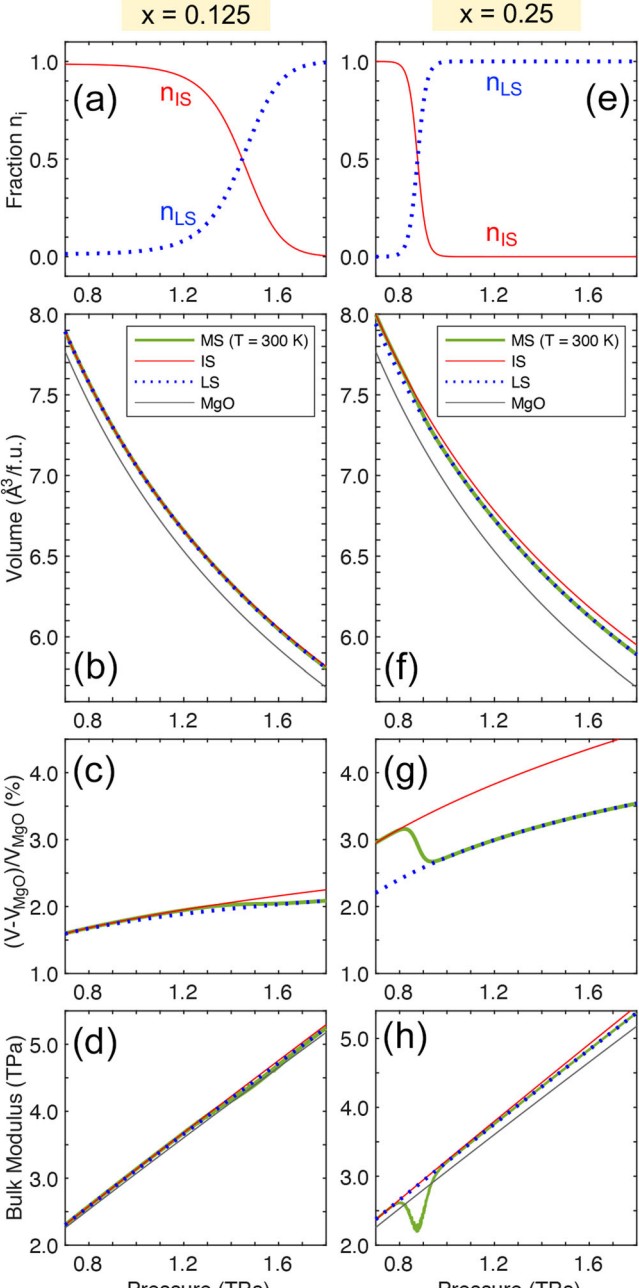

**Fig. 7 Spin transition and accompanying volume/elastic anomalies of (r)B2 $(Mg_{1-x}Fe_x)O$ at room temperature. a–d** rB2 $(Mg_{0.875}Fe_{0.125})O$; **e–h** B2 $(Mg_{0.75}Fe_{0.25})O$. **a, e** Fractions of the intermediate-spin (IS) and low-spin (LS) states; **b, f** compression curves; **c, g** relative volume difference with respect to pure B2 MgO; **d, h** isothermal bulk modulus.

structural, spin, and metal–insulator transition (insulating B1 LS → metallic B2 IS) at 0.539 TPa, followed by a spin transition (metallic B2 IS–LS) at 0.855 TPa. Remarkably, Fe's total electron spin $S$ re-emerges from 0 to 1 in the B1–(r)B2 transition. In addition, structural, spin, and metal–insulator transitions of B2 $(Mg_{1-x}Fe_x)O$ can also be induced by the change of Fe concentration ($x$). These results suggest that Fe and spin transition may greatly affect planetary interiors over a vast pressure range, considering the anomalous changes of elastic, transport, and thermal properties accompanying the spin and/or metal–insulator transition.

## Methods

**Computation**. In this work, all major calculations are performed using the QUANTUM ESPRESSO codes[58]. We use ultrasoft pseudopotentials (USPPs) generated with the Vanderbilt method[59]. The valence electron configurations for the generations are $2s^2 2p^6 3s^2 3p^0 3d^0$, $3s^2 3p^6 3d^6.5 4s^1 4p^0$, and $2s^2 2p^4 3d^0$ for Mg, Fe, and O, respectively; the cutoff radii are 1.4, 1.8, and 1.0 a.u. for Mg, Fe, and O, respectively. The aforementioned USPPs of Mg and O have been used in refs. [15,16], and the USPP of Fe has been used in refs. [22,49–54]. To properly treat the on-site Coulomb interaction of the Fe $3d$ electrons, we adopt the local density approximation + self-consistent Hubbard $U$ (LDA+$U_{sc}$) method, with the Hubbard $U$ parameters computed self-consistently[60–63]. Briefly speaking, we start with an LDA+$U$ calculation with a trial $U$ (the "input $U_{in}$") to obtain the desired spin state for $(Mg_{1−x}Fe_x)O$. For this LDA+$U_{in}$ state, we compute the second derivative of the LDA energy with respect to the $3d$ electron occupation at the Fe site ($d^2 E_{LDA}/dn^2$) via a density functional perturbation theory (DFPT) approach[63] implemented in QUANTUM ESPRESSO. This second derivative, $d^2 E_{LDA}/dn^2$, is considered as the "output $U_{out}$" and will be used as $U_{in}$ in the next iteration. Such a procedure is repeated until self-consistency is achieved, namely, $U_{in} = U_{out} \equiv U_{sc}$. Phonon calculations are performed using the Phonopy code[64], which adopts the finite-displacement (frozen phonon) method. The third-order Birch–Murnaghan equation of state (3rd BM EoS) is used for the EoS fitting.

**Thermodynamic model**. In this work, analysis for the IS–LS transition in (r)B2 $(Mg_{1−x}Fe_x)O$ at room temperature (300 K) is based on the thermodynamic model detailed in Supplementary Note 5. Plotted in Fig. 7, the IS/LS fractions and the EoS of the MS phase are given by Supplementary Eqs. 9 and 10, respectively.

## Data availability

The authors declare that the main data supporting the findings of this study are contained within the paper and its associated Supplementary Information. Example input and output files of our calculations (using QUANTUM ESPRESSO) have been deposited at Zenodo (https://doi.org/10.5281/zenodo.6283200). All other relevant files are available from the corresponding author upon reasonable request.

## Code availability

In this work, all major calculations are performed using the QUANTUM ESPRESSO (QE) codes, and phonon calculations are performed using Phonopy, as described in the Methods Section. Both QE and Phonopy are open-source packages; they can be downloaded for free from http://www.quantum-espresso.org/ and http://phonopy.github.io/phonopy/, respectively. Detailed information for the compilation and installation of these codes are contained in their own websites.

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

## Acknowledgements

H.H. acknowledges support from the Ministry of Science and Technology of Taiwan under Grant Numbers MOST 107-2112-M-008-022-MY3, 107-2119-M-009-009-MY3, and 110-2112-M-008-033. K.U. acknowledges support from the JSPS Kakenhi Grant Numbers 17K05627 and 21K03698. Calculations were performed at National Center for High-performance Computing, Taiwan, and the Global Scientific Information and Computing Center at Tokyo Institute of Technology, Japan.

## Author contributions

Both authors designed and planned this research. K.U. initiated the work by performing preliminary calculations. H.H. performed the major calculations. Both authors analyzed the calculation results. H.H. wrote the manuscript.

## Competing interests

The authors declare no competing interests.
