## [Peer Review File · Nature Communications]

REVIEWER COMMENTS

Reviewer #1 (Remarks to the Author):

The authors presented a first-principles computational study of spin-associated transition of (Mg,Fe)O considering its B1 and B2 phases at pressures up to 1.8 TPa. MgO and its iron alloys are widely studied because of their relevance to Earth and exoplanets. In particular, they predicted a transition of low-spin B1 phase to intermediate-spin B2 phase around 0.6 TPa. It is interesting to see a finite total electron spin of iron ($S = 1$) in the B2 phase before the B2 phase becomes non-magnetic ($S = 0$) upon further compression. The authors also discussed rhombohedral distortions and the metal-insulator transition in the B2 phase.

The predicted behavior of B2 phase of (Mg,Fe)O appears to be strongly sensitive to Fe concentration. The stability field of intermediate spin B2 phase for 25% iron case is much narrower than that for 12.5% iron case. Also, B2 phase is metallic for high iron concentration whereas it is an insulator for low concentration or pure MgO. As shown in Fig. 5, the enthalpy differences between different B2 phases are small, particularly, for intermediate spin state. It is important to assess how sensitive this energetics with respect to different computational factors. For instance, the values of U though obtained in a self-consistent way may not be uniquely defined. So the question is what happens if different U values are used. While LDA was used in this work, GGA is usually considered to be more appropriate for transition metals and iron-bearing systems. Previous studies (e.g., Holmstrom Stixrude, PRL, 2016; Ghosh and Karki, Sci. Rep., 2016) have shown that the predicted high spin to low spin transition of iron in (Mg,Fe)O differs significantly between GGA and LDA with/without the Hubbard term. More tests along this direction will be helpful.

The results analysis and discussion as presented in the paper look mostly fine. However, it is not clear what implications are, particularly, for planetary interiors. Can the authors be more specific in terms of density and composition.

Reviewer #2 (Remarks to the Author):

The manuscript by H. Hsu and K. Umemoto presents a computational study of the (Mg,Fe)O system and discusses, in particular, the magnetic and structural phase transitions this mineral (one of the most abundant in the Earth lower mantle) undergoes under pressure. The range of pressure this work explores (TPa) is actually relevant for terrestrial (rocky) exoplanet (especially those with a larger mass than the Earth and consequently higher pressures in their interior) where this mineral is still expected to be an important one. The study focusses specifically on two compositions of $(M_{1-x}Fe_x)O$: $x = 0.125$ and 0.250 , sufficient to capture the effects of the growing concentration of Fe with increasing depth on the considered structural and elastic properties. For both concentrations the manuscript discusses the behavior of two structural phases, relevant in the considered pressure range: B1 and B2. The authors discuss thoroughly their electronic structure, symmetry and deformations, linking the structural response to pressure to the modifications the ordering of the Fe d states undergo when the crystal field changes. The effects of finite (high) temperatures are partially taken into account in the vicinity of phase transitions through a thermodynamic model based on the entropy of mixing of various phases at different spin (while the vibrational entropy is neglected in this work).

The manuscript is well structured and organized, written in a clear way, easy to read. The physics it discusses is sound and the results are original and quite important for the scientific community.

Authors, who are experts of this system, having studied already the structural/magnetic transition it undergoes in its B1 phase while compressed in the lower Earth's mantle pressure range, report a structural phase transition between B1 and B2 phases (accompanied by a transition from low to intermediate spin for Fe) and a second one in the B2 phase from IS to LS. The first of these is quite remarkable: in fact it makes the system re-acquire a spin ($S = 1$ of IS B2) from a state at $S = 0$ (LS B1), which goes against the tenet that pressure suppresses magnetization. In addition, for higher concentrations of Fe ($x = 0.25$) it also corresponds to an insulator-to-metal transition since the B2 phase is metallic.

Given the quality of the work and the originality of the results I think the paper deserves publication, even in the present form. The only comment I have actually concerns the metallic state of B2. As far as I understand authors judge this phase to be metallic based just on the partial occupation of t_{2g} states (in the undistorted structure) and on the degeneracy of the multiplets resulting from the structural distortions. However, using these arguments is quite dangerous when working with DFT and band theory. In fact, there are many Mott insulators that, due to symmetry, appear to have a metallic band structure, although they are obviously non conductive. Have authors tried to further lower the symmetry of the system (e.g. eliminating all possible residual degeneracies) to check whether the metallic behavior would be robust against it? Although I strongly recommend the authors to take this point into consideration and to possibly update their paper accordingly I leave it as an optional modification and recommend the publication of the manuscript as is.

Reviewer #3 (Remarks to the Author):

The authors conducted first-principles calculations (LDA+U) to investigate the spin states and crystal structures of ferropericlase (Mg,Fe)O at TPa pressure range. They present two major results on two ferropericlase compositions (12.5% and 25% Fe; Fp15 and Fp25): (1). In 12.5% ferropericlase, B1 to rhombohedral B2 phase transition and the intermediate spin to lower spin transition in the rB2 phase; (2). in 25% ferropericlase, B1-B2 transition and the IS-LS spin transition. These results are surprising and warrant a publication in Nature Communications if they can be verified. Like the authors mentioned in the manuscript, the B1-B2 transition is expected to occur at high pressure in MgO-FeO system. However, the rhombohedral B2 phase is totally new. The reemergence of the spin state and the IS-LS transition in the B2 phase are unexpected and have important implications to our understanding of the interiors of exoplanets.

The authors should provide an explanation to why the Fp15 and Fp25 compositions have such a distinct structural transition behavior. The difference in the amount of iron in these two systems is rather small so one would not expect to see such a major difference. MgO-FeO system is a solid solution system so it is just a mystery to me that these two compositions display drastically different phase transition sequences. As shown in Fig 3, the lattice parameters in the rhombohedral B2 change drastically with pressure. Why is that? It will be useful if the authors construct a composition-pressure diagram to depict the phase transition sequence using

literature data. Comparison to literature data in the MgO-FeO system will also help justify their calculations.

Reviewer #1 (Remarks to the Author)

The authors presented a first-principles computational study of spin-associated transition of (Mg,Fe)O considering its B1 and B2 phases at pressures up to 1.8 TPa. MgO and its iron alloys are widely studied because of their relevance to Earth and exoplanets. In particular, they predicted a transition of low-spin B1 phase to intermediate-spin B2 phase around 0.6 TPa. It is interesting to see a finite total electron spin of iron ($S = 1$) in the B2 phase before the B2 phase becomes non-magnetic ($S = 0$) upon further compression. The authors also discussed rhombohedral distortions and the metal-insulator transition in the B2 phase.

Reviewer's Comment

The predicted behavior of B2 phase of (Mg,Fe)O appears to be strongly sensitive to Fe concentration. The stability field of intermediate spin B2 phase for 25% iron case is much narrower than that for 12.5% iron case. Also, B2 phase is metallic for high iron concentration whereas it is an insulator for low concentration or pure MgO. As shown in Fig. 5, the enthalpy differences between different B2 phases are small, particularly, for intermediate spin state. It is important to assess how sensitive this energetics with respect to different computational factors. For instance, *the values of U though obtained in a self-consistent way may not be uniquely defined*. So the question is what happens if different U values are used. While LDA was used in this work, GGA is usually considered to be more appropriate for transition metals and iron-bearing systems. Previous studies (e.g., Holmstrom Stixrude, PRL, 2016; Ghosh and Karki, Sci. Rep., 2016) have shown that the predicted high spin to low spin transition of iron in (Mg,Fe)O differs significantly between GGA and LDA with/out the Hubbard term. More tests along this direction will be helpful.

Author's response

Before proceeding to further discussion, we would like to clarify that we had already performed extensive test calculations on $(\text{Mg}_{1-x}\text{Fe}_x)\text{O}$ with various combinations of the functional (LDA/GGA) and Hubbard U parameter, as requested by the Reviewer. In our previous submission, GGA+ U calculations with $U = 6, 7,$ and 8 eV were already included in Supplementary Information (SI) but mislabeled as LDA+ U . We apologize for such a mistake and the confusion it may have caused. In this revision, the GGA+ U calculations (Fig. S2) have been correctly labeled and also revised by including more data points for the Birch-Murnaghan EoS fitting. Newly added is Fig. S3, which shows additional

LDA+ U calculations with $U = 10$ and 14 eV. The procedure outlined in Figs. S2 and S3 is actually the common practice of DFT+ U calculations: Treating U as a fitting parameter, performing calculations with various trial U and see how the calculation results are affected. Overall, the results obtained from the DFT+ U test calculations with trial U (Figs. S2 and S3) are consistent with the LDA+ U_{sc} results shown in the main text (Fig. 6), as detailed below.

In Fig. S2, GGA+ U calculations for $(\text{Mg}_{1-x}\text{Fe}_x)\text{O}$ with trial $U = 6, 7,$ and 8 eV are shown. For $x = 0.125$ [Figs. S2(a)–S2(c)], a transition from the B1 LS to the rB2 IS state occurs at ~ 0.63 TPa, followed by an IS–LS transition in the rB2 phase. For $x = 0.25$ [Figs. S2(d)–S2(f)], a transition from the B1 LS to the B2 IS state occurs at ~ 0.54 TPa, followed by an IS–LS transition in the B2 phase. Clearly, the predicted B1–(r)B2 structural-transition pressures are barely affected by U . In contrast, the predicted IS–LS spin-transition pressure in the (r)B2 phase significantly increases with U : As U increases from 6 to 8 eV, the IS–LS transition pressure increases from 1.160 to 1.247 TPa for $x = 0.125$ [Figs. S2(a)–S2(c)] and from 0.801 to 0.957 TPa for $x = 0.25$ [Figs. S2(d)–S2(f)].

In Fig. S3, LDA+ U calculations with trial U are shown. Within LDA+ U , an exceptionally large U (> 10 eV) is necessary to stabilize rB2 IS state in the case of $x = 0.125$. When $U = 10$ eV [Fig. S3(a)], the enthalpy crossing of rB2 IS and rB2 LS states occurs at 0.516 TPa, lower than the B1–rB2 transition pressure (0.648 TPa). In other words, a *direct* transition from B1 LS to rB2 LS state occurs at 0.648 TPa *without* going through rB2 IS state. When U increases to 14 eV [Fig. S3(b)], rB2 IS becomes the most favorable state in the region of $0.638 < P < 0.821$ TPa, and the sequence of the transitions becomes B1 LS \rightarrow rB2 IS \rightarrow rB2 LS, same as the GGA+ U results (Fig. S2). For $x = 0.25$ [Figs. S3(c) and S3(d)], the sequence of the transitions remains B1 LS \rightarrow B2 IS \rightarrow B2 LS, regardless of U . Within LDA+ U , the predicted B1–(r)B2 transition pressure is barely affected by U and nearly the same as the GGA+ U results: ~ 0.64 TPa for $x = 0.125$ [Figs. S3(a) and S3(b)] and ~ 0.54 TPa for $x = 0.25$ [Figs. S3(c) and S3(d)]. The predicted IS–LS transition pressure in (r)B2 $(\text{Mg}_{1-x}\text{Fe}_x)\text{O}$ increases with U : From 0.516 to 0.821 TPa for $x = 0.125$; from 0.934 to 1.199 TPa for $x = 0.25$.

The above DFT+ U test calculations clearly indicate that the choice of the functional (LDA/GGA) and U parameter *barely* affects the predicted B1–(r)B2 transition pressure but *significantly* affects the IS–LS transition pressure in the (r)B2 phase. When U increases by 2 and 4 eV in GGA+ U and LDA+ U , respectively, the IS–LS transition pressure in the B2 phase increases by ~ 0.1 TPa (100 GPa) and > 0.2 TPa (200 GPa), respectively. Based on our own experience and the papers mentioned by the Reviewer, similar effects have been observed in other calculations for lower-mantle minerals: Increasing the trial U by 4 eV would increase the spin-transition pressure by *several tens* of GPa. Clearly, spin transition at high pressure cannot be accurately predicted using the trial U approach.

In principle, the Hubbard U parameter, which accounts for the on-site Coulomb interaction, should be computed from the first-principles, and the U value would depend on the iron valence/spin state and pressure. In this work, we compute the U parameter self-consistently (U_{sc}) via a density functional perturbation theory (DFPT) approach (Ref. S6), which is equivalent to the linear response approach that we had used before (Refs. S3–S5). Within this approach, U_{sc} can be determined unambiguously, opposed to the Reviewer’s comment “*the values of U though obtained in a self-consistent way may not be uniquely defined*”. For years, we have extensively tested LDA+ U_{sc} and other DFT+ U methods on Fe-bearing minerals of geophysical and/or geochemical importance, including B1 $(\text{Mg}_{1-x}\text{Fe}_x)\text{O}$ (Ref. 22), Fe-bearing bridgmanite (Refs. 45, 46) and post-

perovskite MgSiO_3 (Refs. 47, 48), ferromagnesite $(\text{Mg}_{1-x}\text{Fe}_x)\text{CO}_3$ (Refs. 49, 51), and the new hexagonal aluminous (NAL) phase (Ref. 50). Throughout these works (Refs. 22, 45–51), we find that spin-transition pressure predicted by $\text{LDA}+U_{\text{sc}}$ is typically within 5–10 GPa around the experimental results, much more accurate than the trial U approach, which has a margin of error of up to tens or hundreds of GPa. The accuracy provided by $\text{LDA}+U_{\text{sc}}$ is of crucial importance for the predictive calculations on Fe-bearing minerals at ultrahigh pressure (TPa) relevant to exoplanet interior but challenging for experiments. We therefore continue adopting $\text{LDA}+U_{\text{sc}}$ in this work.

Comparing the trial $\text{DFT}+U$ (Figs. S2 and S3) and $\text{LDA}+U_{\text{sc}}$ results (Fig. 6), the predicted B1–(r)B2 transition pressures are nearly the same, confirming the conclusion drawn from Figs. S2 and S3: B1–(r)B2 transition is barely affected by the choice of DFT functional and U . As to the IS–LS transition in the (r)B2 phase, the results of $\text{GGA}+U$ with $U = 7$ [Figs. S2(b) and S2(e)] are in best agreement with the $\text{LDA}+U_{\text{sc}}$ results (Fig. 6).

Change of manuscript

Figure S2 is correctly labeled as $\text{GGA}+U$ and has also been revised by including more data points for Birch-Murnaghan EoS fitting. Figure S3 is newly added to show additional $\text{LDA}+U$ results. Discussions for Figs. S2 and S3 (Sec. S3 in SI) have also been rewritten, similar to our response to the Reviewer.

Reviewer's Comment

The results analysis and discussion as presented in the paper look mostly fine. However, it is not clear what implications are, particularly, for planetary interiors. Can the authors be more specific in terms of density and composition.

Author's response

We are glad that the Reviewer considers the *analysis and discussion as presented in the paper look mostly fine*. Below we address the point raised by the Reviewer, about the implications for planetary interiors in terms of volume/density and composition.

(i) *Volume/density* In the main text, we have shown that with Fe concentration $x = 0.25$, spin transition of B2 $(\text{Mg}_{1-x}\text{Fe}_x)\text{O}$ is accompanied by prominent anomalies, including (1) anomalous volume decrease ($\sim 0.5\%$) [Fig. 7(g)], which also indicates anomalous increase of density ρ ($\sim 0.5\%$), and (2) anomalous bulk-modulus softening ($\sim 22\%$) [Fig. 7(h)] (*note*: $K \equiv -V\partial P/\partial V$). While the full elastic tensor C_{ij} and shear modulus G are not computed in this work (these calculations deserve a new paper), anomalies in V (thus ρ) and K shown in Figs. 7(g) and 7(h) clearly indicate anomalous softening in bulk sound velocity $v_\Phi = (K/\rho)^{1/2}$ and compressional wave velocity $v_P = [(K+4G/3)/\rho]^{1/2}$. Furthermore, within the phonon gas model, lattice thermal conductivity $\kappa \approx (1/3)C_V v_P^2 \tau$ (C_V : heat capacity; τ : average phonon scattering time) (Ref. 56), which suggests that anomalous change of v_P may play a role in the anomalous change of thermal conductivity. In fact, anomalous changes of v_Φ , v_P (Refs. 20, 28–33) and κ (Refs. 35, 36) have already been observed in the B1 phase. Therefore, the

volume/elastic anomalies accompanying the spin transition of the B2 phase may be a possible source of seismic and thermal anomalies in exoplanet interiors.

(ii) Composition The interplay between the composition and spin transition can be complicated. The spin transition can be affected by the Fe concentration, and vice versa. For example, in the Earth's mantle condition, Fe partitioning between B1 ($\text{Mg}_{1-x}\text{Fe}_x\text{O}$), Fe-bearing MgSiO_3 bridgmanite, and post-perovskite varies with pressure, temperature, and even the Fe valence/spin state (Refs. 52–54). Therefore, we can expect Fe concentration in B2 ($\text{Mg}_{1-x}\text{Fe}_x\text{O}$) varying with the depth or the interior region in exoplanet interiors, due to the variation of Fe partitioning between B2 ($\text{Mg}_{1-x}\text{Fe}_x\text{O}$) and other mineral phases, including post-perovskite and/or high-pressure silicates (Refs. 15, 16). At the moment, however, the actual mineral phases and Fe partitioning in exoplanet interiors remain unknown. Therefore, we can *at best* discuss the possible effects of composition (Fe concentration) on spin transition as below, instead of the possible effects of spin transition on the composition.

By carefully comparing Figs. 6(a) and 6(b), we can notice that spin, structural, and metal–insulator transition of B2 ($\text{Mg}_{1-x}\text{Fe}_x\text{O}$) can be induced by the change of Fe concentration (x). As can be observed from Fig. 6(a), the most energetically favorable state for ($\text{Mg}_{0.875}\text{Fe}_{0.125}\text{O}$) in the region of 0.642–0.855 TPa is rB2 IS state. In contrast, for ($\text{Mg}_{0.75}\text{Fe}_{0.25}\text{O}$) [Fig. 6(b)], B2 IS state is the most favorable state in the same region (0.642–0.855 TPa). Therefore, in the depth/region with pressure of 0.642–0.855 TPa, if the Fe concentration increases from $x = 0.125$ to 0.25, a simultaneous structural and metal–insulator transition occurs: *insulating* rB2 IS ($\text{Mg}_{0.875}\text{Fe}_{0.125}\text{O}$) \rightarrow *metallic* B2 IS ($\text{Mg}_{0.75}\text{Fe}_{0.25}\text{O}$). Likewise, in the depth/region with pressure of 0.855–1.348 TPa, if x increases from 0.125 to 0.25, a simultaneous structural, spin, and metal–insulator transition occurs: insulating rB2 IS ($\text{Mg}_{0.875}\text{Fe}_{0.125}\text{O}$) \rightarrow metallic B2 LS ($\text{Mg}_{0.75}\text{Fe}_{0.25}\text{O}$). Even in the depth/region of $P > 1.348$ TPa, where only LS Fe^{2+} exists, if x increases from 0.125 to 0.25, a simultaneous structural and metal–insulator transition occurs: insulating rB2 LS ($\text{Mg}_{0.875}\text{Fe}_{0.125}\text{O}$) \rightarrow metallic B2 LS ($\text{Mg}_{0.75}\text{Fe}_{0.25}\text{O}$). On the other hand, if x decreases from 0.25 to 0.125, the aforementioned transitions would be reversed. Noticeably, for the transitions induced by the change of Fe concentration, metal–insulator transition is always included, suggesting significant change of electrical/thermal transport properties of exoplanet interiors due to the variation of Fe partitioning.

Change of manuscript

We have revised our manuscript based on the above discussion. The geophysical (or planetary) implications in terms of volume/density are included in Lines 200–211; the implications in terms of composition are included in Lines 167–185.

Reviewer #2 (Remarks to the Author)

The manuscript by H. Hsu and K. Umemoto presents a computational study of the (Mg,Fe)O system and discusses, in particular, the magnetic and structural phase transitions this mineral (one of the most abundant in the Earth lower mantle) undergoes under pressure. The range of pressure this work explores (TPa) is actually relevant for terrestrial (rocky) exoplanet (especially those with a larger mass than the Earth and consequently higher pressures in their interior) where this mineral is still expected to be an important one.

The study focusses specifically on two compositions of $(M_{1-x}Fe_x)O$: $x = 0.125$ and 0.250 , sufficient to capture the effects of the growing concentration of Fe with increasing depth on the considered structural and elastic properties. For both concentrations the manuscript discusses the behavior of two structural phases, relevant in the considered pressure range: B1 and B2. The authors discuss thoroughly their electronic structure, symmetry and deformations, linking the structural response to pressure to the modifications the ordering of the Fe d states undergo when the crystal field changes. The effects of finite (high) temperatures are partially taken into account in the vicinity of phase transitions through a thermodynamic model based on the entropy of mixing of various phases at different spin (while the vibrational entropy is neglected in this work).

The manuscript is well structured and organized, written in a clear way, easy to read. The physics it discusses is sound and the results are original and quite important for the scientific community.

Authors, who are experts of this system, having studied already the structural/magnetic transition it undergoes in its B1 phase while compressed in the lower Earth's mantle pressure range, report a structural phase transition between B1 and B2 phases (accompanied by a transition from low to intermediate spin for Fe) and a second one in the B2 phase from IS to LS. The first of these is quite remarkable: in fact it makes the system re-acquire a spin ($S = 1$ of IS B2) from a state at $S = 0$ (LS B1), which goes against the tenet that pressure suppresses magnetization. In addition, for higher concentrations of

Fe ($x = 0.25$) it also corresponds to an insulator-to-metal transition since the B2 phase is metallic.

Reviewer's Comment

Given the quality of the work and the originality of the results I think the paper deserves publication, even in the present form. The only comment I have actually concerns the metallic state of B2. As far as I understand authors judge this phase to be metallic based just on the partial occupation of t_{2g} states (in the undistorted structure) and on the degeneracy of the multiplets resulting from the structural distortions. However, using these arguments is quite dangerous when working with DFT and band theory. In fact, there are many Mott insulators that, due to symmetry, appear to have a metallic band structure, although they are obviously non conductive. Have authors tried to further lower the symmetry of the system (e.g. eliminating all passible residual degeneracies) to check whether the metallic behavior would be robust against it?

Although I strongly recommend the authors to take this point into consideration and to possibly update their paper accordingly *I leave it as an optional modification and recommend the publication of the manuscript as is.*

Author's response

We thank the Reviewer for the strong recommendation “*Given the quality of the work and the originality of the results I think the paper deserves publication, even in the present form*”. Throughout the Reviewer’s report, only one point is raised: “*The only comment I have actually concerns the metallic state of B2. ...Have authors tried to further lower the symmetry of the system (e.g. eliminating all passible residual degeneracies) to check whether the metallic behavior would be robust against it?*”. At the end of the report, however, the Reviewer decided to “*leave it as an optional modification and recommend the publication of the manuscript as is*”.

To further improve our manuscript, despite that our response and modification for the manuscript are not required, we address the point raised by the Reviewer and revise our manuscript accordingly. It is well known that B1 (Mg_{1-x}Fe_x)O is insulating in the low Fe concentration regime ($x \leq 0.25$), and the insulating high-spin (HS, $S = 2$) state is achieved via tetragonal Jahn-Teller (J-T) distortion, as further discussed in the next few paragraphs. Intuitively, we had also expected that metallic cubic B2 (Mg_{1-x}Fe_x)O with $x \leq 0.25$ can be further stabilized and become insulating via distortion. Indeed, for $x = 0.125$, rhombohedrally-distorted rB2 (Mg_{0.875}Fe_{0.125})O is insulating, as discussed in Lines 100–118 (Fig. 4) in the main text. Likewise, for cubic B2 (Mg_{0.75}Fe_{0.25})O, we have also lowered its symmetry by applying rhombohedral compression/elongation for the IS/LS state, with the 3d orbital occupations

guided accordingly. To our surprise, the entire crystal and the FeO_8 polyhedra resumed cubic symmetry within a few steps of structural optimization. Consequently, the t_{2g} orbitals are partially occupied, forming a partially-filled t_{2g} band across the Fermi energy, and B2 ($\text{Mg}_{0.75}\text{Fe}_{0.25}$)O remains metallic. We have also tested tetragonal distortion, but after structural optimization, cubic symmetry and thus metallicity persisted. Effects of magnetic ordering were also investigated in these calculations. We find ferromagnetic (FM) state more energetically favorable than antiferromagnetic (AFM) state. In short, metallic cubic B2 ($\text{Mg}_{0.75}\text{Fe}_{0.25}$)O is very robust: No matter what we do, the cubic symmetry and metallicity persist. Descriptions of the aforementioned tests are included in the main text (Lines 125–127 and 60–62).

Next, we further clarify the relation between cubic symmetry and metallicity in the ($\text{Mg}_{1-x}\text{Fe}_x$)O system with $x \leq 0.25$. We fully agree with the Reviewer that standard band theory and standard DFT methods (e.g. LDA or GGA) often fail to open the gap for Mott insulators, as the on-site Coulomb interactions of the localized electrons (e.g. Fe 3d electrons) are not properly treated. To explicitly account for the on-site Coulomb interactions, the Hubbard U correction was proposed (known as the DFT+ U method), as reviewed in Ref. S3 cited in Supplementary Information (SI). Within DFT+ U , the correlation gap of Mott insulators can be opened. A few examples include B1 FeO and NiO, both known to be AFM insulators. As detailed in Ref. S3, GGA fails to open the gap in FeO, while LDA+ U correctly predicts the insulating AFM ground state, with the minority-spin electron fully occupying *one* of the t_{2g} orbitals, despite the cubic symmetry of B1 FeO. Similarly, LDA+ U also properly describes the AFM insulating ground state of cubic B1 NiO. Evidently, DFT+ U can correctly open the gap for Mott insulators, including those with cubic symmetry.

In contrast to cubic B1 FeO, the insulating ground state of B1 ($\text{Mg}_{1-x}\text{Fe}_x$)O with $x \leq 0.25$ is directly related to the tetragonal J-T distortion (first reported in Ref. 25 cited in the main text). At low pressure, B1 ($\text{Mg}_{1-x}\text{Fe}_x$)O with $x \leq 0.25$ is an insulator with HS Fe^{2+} . Both LDA and GGA falsely predict the HS state to be metallic, with cubic symmetry. In the LDA/GGA metallic HS state, all Fe-O bonds of the FeO_6 octahedra have the same length (no J-T distortion), and the degenerate t_{2g} orbitals are partially occupied, forming a partially-filled t_{2g} band. Even within DFT+ U , the correct insulating HS state cannot be obtained automatically, as shown in Fig. R below: LDA+ U calculations ($U = 4$ eV) for B1 ($\text{Mg}_{1-x}\text{Fe}_x$)O with $x = 0.125$ and 0.25 , at volume $17.58 \text{ \AA}^3/\text{f.u.}$ (~ 15 GPa). In one set of the calculations, structural optimizations were performed without breaking the cubic symmetry [Figs. R(a) and R(c)]. For both $x = 0.125$ and 0.25 , the spin-down t_{2g} orbitals are partially occupied by one spin-down electron [inset in Fig. R(a)], and a partially filled t_{2g} band is formed, spanning across the Fermi energy (0 eV) [Figs. R(a) and R(c)]. In another set of the calculations [Figs. R(b) and R(d)], we broke the cubic symmetry by manually imposing tetragonal J-T distortion (stretching the Fe-O bonds on the xy plane), followed by structural optimization. After optimization, the structure remains tetragonal, and the energy is lower than the cubic symmetry. The three t_{2g} orbitals split into to a singlet (d_{xy}) and a doublet ($d_{xz} + d_{yz}$), and the spin-down d_{xy} orbital is occupied by one spin-down electron while the spin-down d_{xz} and d_{yz} orbitals are unoccupied [inset in Fig. R(b)]. Consequently, the spin-down t_{2g} band splits into a completely filled d_{xy} band below the Fermi level (0 eV) and a completely empty $d_{xz} + d_{yz}$ band above 0 eV, as shown in Figs. R(b) and R(d). Evidently, for B1 ($\text{Mg}_{1-x}\text{Fe}_x$)O with $x \leq 0.25$, the *incorrect* metallic HS state (with partially filled t_{2g} band) is a consequence of cubic symmetry, and the *correct* insulating HS state is achieved via J-T distortion.

Figure R. LDA+ U ($U = 4$ eV) density of states (DOS) of B1 HS ($\text{Mg}_{1-x}\text{Fe}_x$)O at $17.58 \text{ \AA}^3/\text{f.u.}$ (~ 15 GPa), with $x = 0.125$ (a,b) and 0.25 (c,d). Without J-T distortion, namely, within cubic symmetry, B1 HS ($\text{Mg}_{1-x}\text{Fe}_x$)O is a metal with a partially-filled t_{2g} band (a,c). With J-T distortion, the t_{2g} band splits into a completely filled d_{xy} band and a completely empty $d_{xz}+d_{yz}$ band in the spin-down channel (b,d).

Similarly, for cubic B2 ($\text{Mg}_{1-x}\text{Fe}_x$)O with $x \leq 0.25$, the crystal field of FeO_8 is in cubic symmetry, and the degenerate t_{2g} orbitals are partially occupied, forming partially-filled t_{2g} bands. Cubic B2 ($\text{Mg}_{0.875}\text{Fe}_{0.125}$)O and ($\text{Mg}_{0.75}\text{Fe}_{0.25}$)O are thus metallic, as detailed in Lines 79–99 (Fig. 3) and Lines 119–141 (Fig. 5), respectively. For $x = 0.125$, B2 ($\text{Mg}_{0.875}\text{Fe}_{0.125}$)O is stabilized via rhombohedral distortion, resulting in insulating rB2 ($\text{Mg}_{0.875}\text{Fe}_{0.125}$)O, as detailed in Lines 100–118 and Fig. 4. For $x = 0.25$, however, the cubic symmetry is very robust (as described in the 2nd paragraph of our response). The main reason is that in B2 ($\text{Mg}_{0.75}\text{Fe}_{0.25}$)O, a 3D network of corner-sharing FeO_8 cubes is formed, which suppress the rhombohedral distortion and further stabilizes the cubic structure, as detailed in Lines 119–141 (Fig. 5).

Change of manuscript

To better explain the metallicity of cubic B2 ($\text{Mg}_{1-x}\text{Fe}_x$)O, we have moved Fig. 3 from SI to the main text to discuss the metallicity of cubic B2 ($\text{Mg}_{0.875}\text{Fe}_{0.125}$)O, see Lines 79–99. We have also rewritten the discussion for B2 ($\text{Mg}_{0.75}\text{Fe}_{0.25}$)O (Fig. 5), see Lines 119–141. Our response to the Reviewer’s comment about lowering the symmetry of cubic B2 ($\text{Mg}_{0.75}\text{Fe}_{0.25}$)O is included in the same part of the revision, see Lines 125–127.

Reviewer #3 (Remarks to the Author)

Reviewer's Comment

The authors conducted first-principles calculations (LDA+U) to investigate the spin states and crystal structures of ferropericlase (Mg,Fe)O at TPa pressure range. They present two major results on two ferropericlase compositions (12.5% and 25% Fe; Fp15 and Fp25): (1). In 12.5% ferropericlase, B1 to rhombohedral B2 phase transition and the intermediate spin to lower spin transition in the rB2 phase; (2). in 25% ferropericlase, B1-B2 transition and the IS-LS spin transition. *These results are surprising and warrant a publication in Nature Communications if they can be verified.* Like the authors mentioned in the manuscript, the B1-B2 transition is expected to occur at high pressure in MgO-FeO system. However, the rhombohedral B2 phase is totally new. The reemergence of the spin state and the IS-LS transition in the B2 phase are unexpected and *have important implications to our understanding of the interiors of exoplanets.*

Author's response

We thank the Reviewer for commenting our manuscript “*surprising and warrant a publication in Nature Communications*” and “*have important implications to our understanding of the interiors of exoplanets*”. In the meantime, we would like to clarify one point: The work presented in this manuscript is a *predictive* calculation. To our knowledge, experiments for the MgO-FeO system *at ultrahigh pressure* have only been conducted on the end members MgO (Refs. 1–4 cited in the main text) and FeO (Refs. 43, 44), not (yet) on Fe-bearing (Mg_{1-x}Fe_x)O with $0 < x \leq 0.25$. Moreover, probing Fe spin states via currently available experimental techniques (e.g. x-ray emission spectroscopy and Mössbauer spectroscopy) *at ultrahigh pressure* has been highly challenging. Therefore, expecting or requesting for experimental verifications at this point would be unrealistic. Nevertheless, as described in Lines 49–57 of the main text, the computational method adopted in this work, LDA+ U_{sc} , has been shown to provide accurate prediction on spin transition of Fe-bearing minerals (Refs. 22, 42–51). Moreover, to examine the robustness of our LDA+ U_{sc} results (Fig. 6), we have also performed GGA+ U calculations ($U = 6, 7,$ and 8 eV) and obtained nearly the same results, as detailed in Sec. S3 (see Fig. S2) of Supplementary Information (SI). Therefore, the main conclusion of this paper, including the *simultaneous* LS-IS and B1-(r)B2 structural transition, and the IS-LS transition in the (r)B2 phase, should be reliable.

Reviewer's Comment

The authors should provide an explanation to why the Fp15 and Fp25 compositions have such a distinct structural transition behavior. The

difference in the amount of iron in these two systems is rather small so one would not expect to see such a major difference. MgO-FeO system is a solid solution system so it is just a mystery to me that these two compositions display drastically different phase transition sequences. As shown in Fig 3, the lattice parameters in the rhombohedral B2 change drastically with pressure. Why is that?

Author's response

This comment from the Reviewer basically concerns the *entire* paper, namely, the unique properties of B2 ($\text{Mg}_{1-x}\text{Fe}_x\text{O}$), which include: (1) The structural and electronic properties are sensitive to the Fe concentration, even at low concentration ($x \leq 0.25$), see Figs. 3–5 in the main text, and (2) the effects of the composition on the structural and spin transitions of ($\text{Mg}_{1-x}\text{Fe}_x\text{O}$), see Fig. 6. Prior to this paper, we had also been expecting the properties of B2 ($\text{Mg}_{1-x}\text{Fe}_x\text{O}$) to remain similar throughout $0 < x \leq 0.25$. To our surprise, properties of B2 ($\text{Mg}_{1-x}\text{Fe}_x\text{O}$) changes drastically as the Fe concentration increases from $x = 0.125$ (referred to as “Fp15” by the Reviewer, although it should be “Fp12”) to $x = 0.25$ (referred to as “Fp25” by the Reviewer). Given this unexpected result, the main focus of this paper is therefore to *analyze, explain, and discuss* the unique properties of the B2 phase in detail. Also, despite the aforementioned composition dependence, both compositions exhibit resemblance in the sequence of transitions: B1 LS \rightarrow rB2 IS \rightarrow rB2 LS for $x = 0.125$, and B1 LS \rightarrow B2 IS \rightarrow B2 LS for $x = 0.25$ (Fig. 6). For some reason, however, these major points in the original manuscript were not successfully conveyed to the Reviewer. Here, we further clarify these points and also revise our manuscript for better clarity, as described below.

Before discussing the newly reported (r)B2 ($\text{Mg}_{1-x}\text{Fe}_x\text{O}$) in this paper, we first use the well-known B1 phase as an example to explain the relation between lattice distortion and insulating state in the ($\text{Mg}_{1-x}\text{Fe}_x\text{O}$) system with $x \leq 0.25$. In the B1 phase, Fe substitute Mg in the 6-coordinate site, forming FeO_6 octahedra, with the t_{2g} orbitals having lower energy than the e_g orbitals, as shown in Fig. S1 of Supplementary Information (SI). At low pressure, B1 ($\text{Mg}_{1-x}\text{Fe}_x\text{O}$) with $x \leq 0.25$ is known to be an insulator with high-spin (HS, $S = 2$) Fe^{2+} . This HS insulating state is stabilized via tetragonal Jahn-Teller (J-T) distortion, with longer Fe-O bonds on the xy plane and *slightly* longer lattice parameters in the x and y direction. Without J-T distortion, the HS state would be metallic and has higher energy. This finding was first reported in Ref. 25 cited in the main text and is also shown in Fig. R below: LDA+ U calculations ($U = 4$ eV) for B1 ($\text{Mg}_{1-x}\text{Fe}_x\text{O}$) with $x = 0.125$ and 0.25 at volume $17.58 \text{ \AA}^3/\text{f.u.}$ (~ 15 GPa). In one set of the calculations, structural optimizations were performed *without* manually breaking the cubic symmetry [Figs. R(a) and R(c)]. Consequently, within cubic symmetry, the t_{2g} and e_g orbitals are three and two-fold degenerate, respectively [inset in Fig. R(a)]. For the HS state, the spin-down t_{2g} orbitals are partially occupied by one spin-down electron [inset in Fig. R(a)], forming a partially filled t_{2g} band spanning across the Fermi energy (set at 0 eV) [Figs. R(a) and R(c)]. In another set of the calculations, we broke the cubic symmetry by manually imposing tetragonal J-T distortion (stretching the Fe-O bonds on the xy plane), followed by structural optimization [Figs. R(b) and R(d)]. After structural optimization, the structure remains tetragonal, and the energy is lower than the cubic symmetry. With J-T distortion, the three t_{2g} orbitals split into a singlet (d_{xy}) with lower energy and a

Figure R. LDA+ U ($U = 4$ eV) density of states (DOS) of B1 HS ($\text{Mg}_{1-x}\text{Fe}_x$)O at $17.58 \text{ \AA}^3/\text{f.u.}$ (~ 15 GPa), with $x = 0.125$ (a,b) and 0.25 (c,d). Without J-T distortion, namely, within cubic symmetry, B1 HS ($\text{Mg}_{1-x}\text{Fe}_x$)O is a metal with a partially-filled t_{2g} band (a,c). With J-T distortion, the t_{2g} band splits into a completely filled d_{xy} band and a completely empty $d_{xz}+d_{yz}$ band in the spin-down channel (b,d).

doublet (d_{xz} and d_{yz}) with higher energy [inset in Fig. R(b)]. For the HS state, the spin-down d_{xy} orbital is occupied by one spin-down electron, and the spin-down d_{xz} and d_{yz} orbitals are unoccupied [inset in Fig. R(b)]. Consequently, the spin-down t_{2g} band splits into a completely filled d_{xy} band and a completely empty $d_{xz}+d_{yz}$ band, as shown in Figs. R(b) and R(d). Evidently, for B1 ($\text{Mg}_{1-x}\text{Fe}_x$)O with $x \leq 0.25$, the *incorrect* metallic HS state is a consequence of cubic symmetry, and the *correct* insulating HS state is achieved via distortion.

Similarly, *cubic* B2 ($\text{Mg}_{1-x}\text{Fe}_x$)O with $x \leq 0.25$ is also metallic, with a partially-filled t_{2g} band. To achieve an insulating state in this phase, lattice distortion would be necessary. Indeed, for $x = 0.125$, we find *cubic* B2 ($\text{Mg}_{0.875}\text{Fe}_{0.125}$)O *dynamically unstable*, and the IS/LS state is stabilized via rhombohedral compression/elongation, referred to as rB2 IS/LS ($\text{Mg}_{0.875}\text{Fe}_{0.125}$)O, as detailed in Lines 79–99 (Fig. 3) and 100–118 (Fig. 4) in the main text. (*Note:* Figure 3 was formerly placed in SI of the original manuscript.) For $x = 0.25$, to our surprise, the cubic structure is very robust: When we applied rhombohedral distortion on B2 ($\text{Mg}_{0.75}\text{Fe}_{0.25}$)O, the crystal structure resumed cubic symmetry within a few steps of the structural optimization. Furthermore, *cubic* B2 ($\text{Mg}_{0.75}\text{Fe}_{0.25}$)O is *dynamically stable*. The main reason is that in B2 ($\text{Mg}_{0.75}\text{Fe}_{0.25}$)O, a three-dimensional (3D) network of corner-sharing FeO_8 is formed, which suppresses the rhombohedral distortion and further stabilizes the cubic structure, as detailed in Lines 119–141 (Fig. 5). Within the cubic structure, B2 ($\text{Mg}_{0.75}\text{Fe}_{0.25}$)O is metallic,

regardless the spin state. For the Reviewer's convenience, these results are further described below (but briefer than the manuscript).

In the B2 phase, Fe substitutes Mg in the 8-coordinate site, forming FeO₈ polyhedra, where the t_{2g} orbitals have *higher* energy than the e_g orbitals (opposite to the B1 phase). For cubic B2 (Mg_{0.875}Fe_{0.125})O, the FeO₈ polyhedra are in cubic (O_h) symmetry [Figs. 3(a) and 3(f)], and the t_{2g} and e_g orbitals are three and two-fold degenerate, respectively [Figs. 3(c) and 3(h)]. For the IS and LS states, the t_{2g} orbitals are partially occupied [Figs. 3(c) and 3(h)], forming partially-filled t_{2g} bands [Figs. 3(e) and 3(j)]. Therefore, cubic B2 (Mg_{0.875}Fe_{0.125})O is metallic, regardless of the spin state. However, cubic B2 (Mg_{0.875}Fe_{0.125})O is *dynamically unstable* [see negative frequencies in Figs. 3(b) and 3(g)]. To stabilize B2 (Mg_{0.875}Fe_{0.125})O, the FeO₈ cubes and *the entire crystal structure* undergo rhombohedral distortion [referred to as rB2 (Mg_{0.875}Fe_{0.125})O, see Fig. 4]. (*Note*: Figure 4 in the revised manuscript was formerly Fig. 3 in the original manuscript.) Briefly speaking, the IS/LS state is stabilized via rhombohedral compression/elongation, with shortened/stretched Fe-O bonds along the [111] direction, and the rhombohedral angle (α) greater/smaller than 90° [Figs. 4(a) and 4(f)]. In the crystal field with rhombohedral compression/elongation, the t_{2g} orbitals split into a singlet a_{1g} and a doublet e'_g , and the a_{1g} orbital has higher/lower energy than the e'_g orbitals, due to the shortened/stretched Fe-O bonds along the [111] direction [Figs. 4(c) and 4(h)]. For the IS state, the spin-up e'_g orbitals are completely occupied by two spin-up electrons, and the spin-up a_{1g} orbital is completely unoccupied [Fig. 4(c)]. For the LS state, the a_{1g} orbital is completely occupied, and the e'_g orbitals are completely unoccupied [Fig. 4(h)]. Consequently, a gap is opened between the a_{1g} and e'_g bands [Figs. 4(e) and 4(j)]. Clearly, the insulating state of rB2 (Mg_{0.875}Fe_{0.125})O is a direct consequence of rhombohedral distortion, similar to the insulating HS state in the B1 phase achieved via tetragonal J-T distortion (Fig. R).

To our surprise, in B2 (Mg_{0.75}Fe_{0.25})O (namely, $x = 0.25$), rhombohedral distortion does not occur, and B2 (Mg_{0.75}Fe_{0.25})O remains cubic (see Fig. 5). Briefly speaking, with $x = 0.25$, the Fe concentration is high enough to form a 3D network of corner-sharing FeO₈ cubes; in contrast, for $x \leq 0.125$, the FeO₈ polyhedra are isolated (unconnected), as shown in Fig. 1 (bottom panels). For isolated FeO₈ polyhedra, rhombohedral distortion is allowed and favored, as observed in rB2 (Mg_{0.875}Fe_{0.125})O (Fig. 4). The connectivity of the 3D FeO₈ network in B2 (Mg_{0.75}Fe_{0.25})O, however, suppresses the rhombohedral distortion and further stabilizes the cubic structure. When performing structural optimization, we did apply rhombohedral distortion on B2 (Mg_{0.75}Fe_{0.25})O, but within a few steps, the crystal structure and FeO₈ polyhedra resumed cubic symmetry. Based on these tests, *dynamical stability* of cubic B2 (Mg_{0.75}Fe_{0.25})O can thus be expected [Fig. 5(b)–5(d)]. Within cubic symmetry, B2 (Mg_{0.75}Fe_{0.25})O is a metal with a partially-filled t_{2g} band [Figs. 5(e)–5(g)], similar to cubic B2 (Mg_{0.875}Fe_{0.125})O [Figs. 3(e) and 3(j)].

Finally, the Reviewer seemed to misunderstand Fig. 4 (formerly Fig. 3 in the original manuscript). In Fig. 4, we plot the atomic structure of rB2 (Mg_{0.875}Fe_{0.125})O at volume 55.310 Å³/f.u. (~1.07 TPa), simply to point out that the IS and LS states are stabilized via rhombohedral compression and elongation, respectively, and therefore, have different structural properties, as explained in the 4th paragraph of our response and also detailed in Lines 100–118. Figure 4 has nothing to do with spin transition. Spin transition is determined by comparing the enthalpy of these two states, as shown in Fig. 6(a): A crossing occurs at 1.438 TPa, indicating a transition from rB2 IS to rB2 LS state. Given

that rB2 IS and LS states have slightly different structures, this transition is in fact a *simultaneous* spin and structural transition: rhombohedrally-compressed IS state ($\alpha > 90^\circ$) \rightarrow rhombohedrally-elongated LS state ($\alpha < 90^\circ$). It is well known that spin transition is often accompanied by a change of structure, as different spin states have different lattice distortions (due to different orbital occupations).

Change of manuscript

To better explain the metallicity of cubic B2 ($\text{Mg}_{1-x}\text{Fe}_x$)O, we have moved Fig. 3 from SI to the main text to discuss the metallicity of cubic B2 ($\text{Mg}_{0.875}\text{Fe}_{0.125}$)O, see Lines 79–99. We have also rewritten the discussion for B2 ($\text{Mg}_{0.75}\text{Fe}_{0.25}$)O (Fig. 5), see Lines 119–141.

Reviewer's Comment

It will be useful if the authors construct a **composition-pressure diagram** to depict the phase transition sequence using literature data. Comparison to literature data in the MgO–FeO system will also help justify their calculations.

Author's response

As mentioned in our response to the Reviewer's first comment, to our best knowledge, there is no experimental data in the literature to make a **composition-pressure diagram** for the MgO–FeO system *at ultrahigh pressure*. Currently, ultrahigh-pressure experiments for the MgO–FeO system have only been conducted on MgO (Refs. 1–4) and FeO (Refs. 43, 44). As described in the main text, experiments have shown that the B1–B2 transition of MgO and FeO occurs at ~ 0.5 TPa (see Line 23) and ~ 0.25 TPa (see Line 44), respectively. In our calculation, we find the B1–B2 transition of MgO occurs at 0.535 TPa, consistent with previous calculations (Refs. 3–14). As the Fe concentration increases from $x = 0$ (MgO) to $x = 0.125$, the B1–B2 transition pressure increases to 0.642 TPa [Fig. 6(a)]. Remarkably, when the Fe concentration further increases to $x = 0.25$, the B1–B2 transition pressure decreases to 0.539 TPa [Fig. 6(b)], indicating a trend that the B1–B2 transition pressure decreases with Fe concentration for $x > 0.125$. This result is consistent with experiments: a much lower B1–B2 transition pressure in FeO (~ 0.25 TPa) than in MgO (~ 0.5 TPa).

Change of manuscript

The above discussion is now included in the manuscript, see Lines 157–163 in the main text.

REVIEWERS' COMMENTS

Reviewer #1 (Remarks to the Author):

The revision of the paper by the authors in response to my review and other two reviews look good. No further comments.

Reviewer #2 (Remarks to the Author):

The authors have made a significant effort to improve their manuscript according to the comments of the referees.

Although I am mostly satisfied with their reply to the question about the metallic character of the IS B2 phase

of the $x=0.75$ Fe-Mg oxide (presumably related to the fact that FeO₈ octahedra share corners when Fe concentration

is higher and the B2 phase stabilized), a small doubt remains about it.

In fact, in order to lower the symmetry of the crystal and check the existence of insulating ground states authors

have tried to distort the same unit cell they have been using through their work for that composition (typically a quite symmetric supercell of the reference crystal). However,

as illustrated for example in Physics Letters A 374, 3793–3796 (2010) or in PHYSICAL REVIEW B 102, 045112 (2020),

sometimes it takes a supercell of the reference crystal for the electronic system to achieve sufficiently low symmetry and to stabilize an insulating ground state. So a chance exists in my opinion that the B2 phase is insulating

within some non trivial long-range charge- or orbital-ordered ground state.

In any case this problem would deserve a separate investigation of its own (outside the scope of the manuscript) while leaving other results (e.g., the spin "resurrection from LS to IS) largely unaffected.

Therefore, also in consideration of the fact that I had left this point as optional, I recommend publication of the revised manuscript in its present form.

Reviewer #3 (Remarks to the Author):

I applaud the authors to do such a wonderful job in responding to both reviewers' comments. The responses and the revised manuscript read very well. I understand that these took the authors quite a lot of time to do so, but this is worth it. I recommend publication of the paper as is.

Reviewer #1 (Remarks to the Author)

The revision of the paper by the authors in response to my review and other two reviews look good. No further comments.

Author's response

We thank the Reviewer for the positive comments and the recommendation.

Change of manuscript

Based on the Reviewer's comments, no change is necessary.

Reviewer #2 (Remarks to the Author)

The authors have made a significant effort to improve their manuscript according to the comments of the referees.

Although I am mostly satisfied with their reply to the question about the metallic character of the IS B2 phase of the $x=0.75$ Fe-Mg oxide (presumably related to the fact that FeO₈ octahedra share corners when Fe concentration is higher and the B2 phase stabilized), a small doubt remains about it.

In fact, in order to lower the symmetry of the crystal and check the existence of insulating ground states authors have tried to distort the same unit cell they have been using through their work for that composition (typically a quite symmetric supercell of the reference crystal). However, as illustrated for example in Physics Letters A 374, 3793–3796 (2010) or in PHYSICAL REVIEW B 102, 045112 (2020), sometimes it takes a supercell of the reference crystal for the electronic system to achieve sufficiently low symmetry and to stabilize an insulating ground state. So a chance exists in my opinion that the B2 phase is insulating within some non trivial long-range charge- or orbital-ordered ground state.

In any case this problem would deserve a separate investigation of its own (outside the scope of the manuscript) while leaving other results (e.g., the spin "resurrection from LS to IS) largely unaffected.

Therefore, also in consideration of the fact that I had left this point as optional, I recommend publication of the revised manuscript in its present form.

Author's response

We thank the Reviewer for the positive comments and the recommendation.

We also appreciate the Reviewer for raising an open question, which showcases the complicated nature of B2 ($\text{Mg}_{0.75}\text{Fe}_{0.25}\text{O}$): Could the ground state of B2 ($\text{Mg}_{0.75}\text{Fe}_{0.25}\text{O}$) be an orbital-ordered (OO) insulating state, rather than the dynamically stable metallic state reported in this paper? At the moment, we decide not to launch an investigation for this question, as described below.

(1) We fully agree with the Reviewer that an OO insulating state for B2 ($\text{Mg}_{0.75}\text{Fe}_{0.25}\text{O}$) could be possible. Just as the Reviewer has commented, however, “*a separate investigation... (out of the scope of the manuscript)*” is needed. At the moment, no one really knows if such a state could be obtained. If not, then the metallic state reported in this paper is the correct ground state. Moreover, even if an OO insulating state could be obtained, its energy/enthalpy is not necessarily lower than the metallic state reported in this paper. Namely, the currently reported metallic state is still more energetically favorable and thus dominates the interior of terrestrial exoplanets.

(2) Consider, hypothetically, that an OO insulating state is obtained and is more favorable than the reported metallic state. Elevated temperature, however, can easily destroy the long-range orbital ordering. Therefore, in exoplanet interiors, the reported metallic phase of B2 ($\text{Mg}_{0.75}\text{Fe}_{0.25}\text{O}$) would coexist (if not dominate) with the OO insulating phase and still play a significant role.

Given the above factors, we decide not to further investigate the possibility of an OO insulating state at the moment. Either way, metallic B2 ($\text{Mg}_{0.75}\text{Fe}_{0.25}\text{O}$) would be of importance for exoplanet interiors, and its properties should be investigated and reported, as we did in this paper.

Change of manuscript

Based on the Reviewer’s comments, no change is necessary.

Reviewer #3 (Remarks to the Author)

I applaud the authors to do such a wonderful job in responding to both reviewers' comments. The responses and the revised manuscript read very well. I understand that these took the authors quite a lot of time to do so, but this is worth it. I recommend publication of the paper as is.

Author’s response

We thank the Reviewer for the positive comments and the recommendation.

Change of manuscript

Based on the Reviewer’s comments, no change is necessary.